# Diagnostic Management of Acute Pulmonary Embolism in COVID-19 and Other Special Patient Populations

**DOI:** 10.3390/diagnostics12061350

**Published:** 2022-05-30

**Authors:** Emily S. L. Martens, Menno V. Huisman, Frederikus A. Klok

**Affiliations:** Department of Medicine–Thrombosis and Haemostasis, Leiden University Medical Centre, Albinusdreef 2, 2333 ZA Leiden, The Netherlands; e.s.l.martens@lumc.nl (E.S.L.M.); m.v.huisman@lumc.nl (M.V.H.)

**Keywords:** venous thromboembolism, pulmonary embolism, COVID-19, diagnosis, D-dimer, computed tomography

## Abstract

Venous thromboembolism (VTE), in particular acute pulmonary embolism (PE), has been shown to be a frequent and potentially fatal complication of coronavirus disease 2019 (COVID-19). In response to the observed thrombotic complications, a large number of studies has been devoted to the understanding and management of COVID-19-associated coagulopathy. Notably, only a limited number of mostly retrospective studies has focused on the optimal diagnostic strategy for suspected PE in COVID-19 patients. As in other special populations, the accuracy of diagnostic algorithms for PE-exclusion has been debated in this specific patient subgroup as the specificity of D-dimer assays and clinical decision rules (CDRs) may be lower than normal. From this viewpoint, we discuss the current state-of-the-art diagnostic algorithms for acute PE with a focus on patients with COVID-19 in the perspective of other special patient populations. Furthermore, we summarize current knowledge regarding the natural history of PE resolution with anticoagulant treatment in patients with COVID-19.

## 1. Introduction

Diagnosing PE is long recognized to be difficult as the clinical signs and symptoms of acute PE are non-specific and show overlap with many other acute cardiopulmonary conditions [1]. The diagnosis of PE therefore relies on targeted tests to establish the pre-test probability of PE using validated CDRs, D-dimer testing and imaging techniques of which computed tomography pulmonary angiography (CTPA) is the current standard [1]. Notably, it has been debated whether these algorithms are applicable to specific patient subgroups such as pregnant women, patients with cancer or hospitalized patients, as the specificity of the D-dimer and CDR may be less in them [2,3]. More recently, hospitals were faced with large numbers of respiratory compromised patients with COVID-19 pneumonitis. These latter patients were found to have a high incidence of thrombotic complications, in particular acute PE [4,5,6,7,8,9,10,11,12]. Whether the usual diagnostic algorithms for suspected acute PE could be used in this setting, with known strong inflammatory response leading to (very) high D-dimer levels, healthcare resources being under pressure and strict isolation measures slowing down routine medical procedures, is not well established. In this review, we discuss the current state-of-the art of diagnosing acute PE with a special focus on special populations including patients with COVID-19. For the latter subgroup, we performed an extensive literature search (Appendix A) on the published literature regarding the safety and efficiency of the validated diagnostic algorithms for suspected PE.

## 2. Diagnostic Algorithms for Acute PE

Once acute PE is considered, many patients are referred for diagnostic imaging since that is the only way to confirm the diagnosis [13]. However, imaging tests such as CTPA are time-consuming and associated with healthcare costs, radiation exposure, contrast-induced nephropathy and allergic reactions [14,15,16,17]. Moreover, due to substantial improvements in imaging techniques, more patients may be exposed to potentially dangerous anticoagulant treatment as a consequence of the increasing number of isolated subsegmental PE diagnoses [18,19,20]. Therefore, various diagnostic algorithms of non-invasive tests have been developed and validated, aiming to safely exclude PE without the need for imaging in all patients. These algorithms start with a CDR and are followed by D-dimer testing in case of a non-high pre-test probability. Following these algorithms, imaging is only indicated in the case of a high pre-test probability and/or a validated abnormal D-dimer result [21,22,23,24,25,26]. While the conventional D-dimer threshold for ruling out acute PE was set at 500 ng/mL, more recent studies have shown that an age dependent or pre-test probability dependent threshold is safe as well and associated with a higher specificity, i.e., more patients can be managed without CTPA [27,28,29,30]. Specifically, the YEARS algorithm involves only three items and a binary D-dimer threshold adapted to the clinical pre-test probability in all patients, allowing PE to be ruled out without imaging in 50% of patients [28]. As a consequence of its simplicity and prove of efficiency, the YEARS algorithm is very compatible with the demands of clinical practice, and is associated with less subsegmental PE diagnoses and considerable cost saving [31,32].

## 3. Diagnosis of Acute PE in Special Populations

The optimal diagnostic workup of suspected PE in special populations remains a point of debate. It is recognized that CDRs and D-dimer tests may not be as accurate in these subgroups compared to the general population. For instance, D-dimer tests often yield false-positive results during pregnancy [33,34,35,36]. Nevertheless, results of a multinational prospective management study of 441 pregnant women with clinically suspected PE suggest that a negative conventional D-dimer result together with a non-high pre-test probability (revised Geneva) score safely excludes PE without the need for CTPA in 12% of patients [34]. As D-dimer levels increase physiologically throughout pregnancy, adjusted trimester-specific D-dimer thresholds or pre-test adjusted D-dimer thresholds may favour the utility of D-dimer testing [35,37]. The latter was tested in the prospective Artemis study in 498 pregnant women with PE-suspicion. The study showed that a pre-test adjusted D-dimer according to YEARS was of significant value across all trimesters of pregnancy by reducing the need for CTPA in 32 to 65% of women, depending on the trimester. Only one woman, in whom PE was ruled out at baseline, developed a popliteal deep vein thrombosis (0.21%, 95% CI 0.04–1.2) during follow-up [29].

Similar to pregnant women, D-dimer levels tend to be higher in cancer patients compared to the general population due to coagulation and fibrinolytic activities contributed by the cancer itself, as well as by cancer treatments [38,39,40]. Furthermore, cancer patients are significantly less likely to be categorized as a low risk according to different CDRs. This is because the majority of the CDRs contain the item ‘active cancer’ [41,42]. Randomized controlled trials evaluating the efficiency and safety of a diagnostic PE algorithm in patients with an active malignancy are currently lacking. The ongoing Hydra study, a prospective diagnostic management study in cancer patients evaluating the safety and efficiency of the YEARS algorithm versus CTPA alone, will hopefully provide more insight in the best diagnostic strategy for diagnosing PE in the latter (NTR: Trial NL7752).

A third category of patients where use of CDRs is much debated concerns hospitalized patients with suspected PE. These patients typically reflect more severe and progressive illness with a high VTE risk. As recently highlighted by an individual participant data meta-analysis (IPD-MA), including over 35,000 patients suspected of PE across different healthcare settings, it is relevant to take PE-prevalence in consideration when judging the performance of the different diagnostic strategies in settings with more high-risk patients [43]. The IPD-MA showed that both failure rate (i.e., false negative result of test) and efficiency became poorer for all currently available diagnostic strategies. In support of this prevalence-adjusted approach, one study, including data obtained from two multicentre prospective management studies (the Christopher study [21] and the Prometheus study [23]) with outpatients and hospitalized patients with suspected PE, demonstrated that the standard diagnostic algorithm with a fixed D-dimer threshold seemed to be the optimal strategy for hospitalized patients [44]. With the lack of a properly powered study in this patient category, it will remain unknown whether modern algorithms such as YEARS are appropriate for hospitalized patients.

## 4. Diagnosis of Acute PE in Patients with COVID-19

It has been questioned whether the usual diagnostic algorithms would be applicable to patients with COVID-19. As with the other special populations as described above, D-dimer levels are often elevated in patients with COVID-19 due to the strong inflammatory response and COVID-19 specific coagulopathy, suggesting a decreased specificity [45]. Moreover, because many of the symptoms of COVID-19 overlap with those of acute PE, e.g., dyspnoea, hypoxia, chest pain and tachycardia, the usual definition of suspected PE may not apply and CDRs may be less accurate. On the other hand, limiting the required number of imaging tests is at least as relevant as in the non-COVID-19 setting. Apart from the usual disadvantages, imaging of this latter category of patients is even more demanding because of logistic strains as time-consuming precautionary measures preventing dissemination, limited resources and radiologic service.

Several studies have suggested higher D-dimer thresholds as a stand-alone test for selecting COVID-19 patients for CTPA [46,47,48]. However, these studies usually establish this threshold based on the optimal cut-off between sensitivity and specificity on the ROC curve, generally resulting in unacceptable proportion of missed PE due to suboptimal values for both sensitivity as specificity. Notably, none of these thresholds have been studied prospectively and external validation is lacking for the vast majority. Accordingly, D-dimer tests should only be used concomitant to CDRs.

Our literature search identified only three relevant studies evaluating the performance of diagnostic algorithms in patients with (suspected) COVID-19 [49,50,51]. One single-centre retrospective study compared the diagnostic accuracy of the widely used Wells and Geneva scores combined with a fixed D-dimer threshold of 500 ng/mL and three alternative approaches, i.e., age-adjusted D-dimer cut-off and the pre-test probability dependent threshold according to YEARS and PEGeD [49]. A total of 300 COVID-19 patients, who were admitted to the emergency department because of PE-suspicion, were analysed. All patients were subjected to D-dimer testing, as well as CTPA. The pre-test probability was retrospectively confirmed in all patients. If there was no documentation for a component of any score, it was considered absent. Moreover, PE was considered equally likely based on the attendant physician’s impression recorded in the medical chart. Through CTPA, PE was confirmed in 45 patients (15%). Following the Wells and revised Geneva score with a fixed D-dimer threshold, PE could have been ruled out in 23 patients, at a failure rate of 8.7%. If these diagnostic algorithms were combined with an aged-adjusted cut-off, PE could have been ruled out in 44 patients at a higher failure rate of 11%. Following the YEARS and PEGeD algorithm, PE could have been ruled out in 85 and 83 patients respectively, at a failure rate of 7.1% and 8.1%. Compared to the other CDRs, combined with a fixed or age-adjusted D-dimer cut-off, application of the YEARS and PEGeD algorithm would have lowered the need for CTPA by 19%.

A prospective observational study covered a cohort of 706 COVID-19 suspected patients admitted to the ED from a large Dutch hospital. All patients were systematically screened for PE using the YEARS algorithm. A definite COVID-19 diagnosis could only be established after full diagnostic work-up including CTPA, since either a positive SARS-CoV-2 RT-PCR or a CO-RADS 4 or 5 could confirm the diagnosis. Following the YEARS algorithm, PE was considered ruled out (without CTPA) in 273 out of 706 patients (39%). A limitation of this study is the lack of follow-up; an assessment of the safety of the YEARS algorithm was therefore not possible.

Up until now, only one prospective diagnostic management study has been published, evaluating the safety and efficiency of validated diagnostic strategies among 707 patients with both (suspected) COVID-19 and suspected acute (first or recurrent) PE [51]. Outpatients and inpatients (ward and ICU) from 14 Dutch hospitals with suspected PE were managed according to the YEARS algorithm (36% of the population, *n* = 255), Wells rule in conjunction with either a fixed or age-adjusted D-dimer threshold (4.2% of the population, *n* = 30) or CTPA directly (52% of the population, *n* = 370) based on local hospital protocol and clinical judgement. Reasons for suspicion of PE included new onset or worsening of chest pain or dyspnoea, new/unexplained tachycardia, a fall in blood pressure not attributable to tachyarrhythmia, hypovolemia, electrocardiogram changes suggestive of PE and increasing D-dimer levels over time. Overall, PE was detected at baseline in 197 patients (28%). Those in whom PE was considered ruled out at baseline and who did not receive therapeutic anticoagulation were followed for 3 months. Results of this prospective study underline the applicability of the YEARS algorithm in patients both suspected of COVID-19 and acute PE: CTPA was avoided in 29% of patients, at a low 1.4% diagnostic failure rate (95% CI 0.04–7.8). In patients in whom the Wells rule was used, CTPA was avoided in only 6.7%, at a higher failure rate (4.3%; 95% CI 0.11–22). The failure rate after a negative CTPA, used as a sole test or within the YEARS, was 3.6% (95% CI 1.6–7.0) and 8.8% respectively.

Overall, the limited available evidence suggests that usual diagnostic algorithms, i.e., YEARS, can be used in patients with COVID-19, although the number of patients who can be managed without CTPA is lower than in the non-COVID-19 population and the failure rate is higher than in non-COVID-19 patients.

## 5. Follow-Up Imaging after Confirmed COVID-19 Associated PE

Notably, the failure rate of a normal CTPA in COVID-19 patients is higher than can be expected in non-COVID-19 patients, underlining the high risk of thrombotic complications in COVID-19 patients, as well as suggesting a low threshold for repeating imaging tests in patients with persistent or worsening cardiopulmonary compromise [52]. Moreover, the rate of recurrent or progressive PE despite anticoagulant therapy is considerable too [53]. However, to date, little is known regarding the natural history of PE resolution with anticoagulant treatment in patients with COVID-19, making the diagnosis of recurrent PE very complex. The resolution rate of COVID-19 patients may also be very relevant for long-term complications such as chronic thromboembolic pulmonary hypertension (CTEPH) as well [54].

Prior to the COVID-pandemic, a Dutch cohort study demonstrated that 84% of patients show complete resolution of PE on repeat CTPA 6 months after the index CTPA [55]. For those experiencing residual thrombotic obstruction, the median obstruction index was limited with a value of 5.0%; at baseline, the obstruction index was 27.5%.

Up until now, only three studies evaluated the thrombus response to anticoagulation treatment in patients with COVID-associated PE subjected to follow-up imaging. The first was a retrospective study in 47 hospitalized patients with COVID-19 and acute PE [56]. Indications for repeat imaging included most frequently shortness of breath (30%) and follow-up after initial PE diagnosis (26%) [56]. After a mean period of 48 days ± 43 days (range, 6–239 days), complete resolution was seen in 72%. Four patients (8.5%) had a new or increased thrombus burden despite anticoagulant therapy, identified on days 9, 20, 41 and 121 of follow-up; two individuals had Apixaban breakthrough, others received subtherapeutic coumadin therapy. Notably, the group of residual thrombus was not further specified in the context of decrease or stable situation of thrombus burden. In the second retrospective study, 6 out of 23 consecutive patients with COVID-19 were subjected to a follow-up CTPA scan because of clinical worsening within 12–28 days after PE diagnosis was confirmed [57,58]. All patients showed residual thrombus obstruction, although with substantial decrease of thrombus load. In the third study, a prospective cohort study, 20 of 77 hospitalized COVID-19 patients with confirmed PE underwent follow-up CTPA (median 105 days (IQR 84–120)) as part of a structured follow-up protocol for severely affected COVID-19 patients [59,60]. Complete thrombus resolution was observed in 60% of the cohort. Six (30%) patients had residual thrombosis, of which two showed a reduction in thrombus load.

These three studies show a somewhat lower rate of clot resolution than in non-COVID-19 PE patients, although the quality of evidence is poor. The earlier and heterogenous timing of follow-up imaging and differences in quantifying the degree of thrombus resolution may further explain between study differences. It is likely that further thrombus resolution occurs beyond the first weeks of treatment. We did not identify studies describing the diagnostic work-up of suspected recurrent COVID-19 associated PE, so the clinical relevance of persistent clots remains unknown. Also, the incidence of chronic thromboembolic pulmonary hypertension (CTEPH) after COVID-19 associated PE, estimated to be 2–3% in non-COVID-19 PE survivors, is currently unknown [61,62,63]. Future studies are required to elucidate the clinical relevance of persistent clots on the long-term outcome of COVID-19 patients with PE.

## 6. Conclusions

In this narrative review we discussed the current state-of-the-art with regard to the application of diagnostic algorithms for suspected PE in COVID-19 patients. As we know from other special populations, in which D-dimer levels tend to be higher and decision rules less accurate, the current diagnostic algorithms still allow us to avoid imaging in a considerable number of patients. For COVID-19, this was 29%. Hence, we strongly support the use of the standard diagnostic algorithms for all patients with suspected PE and specifically for COVID-19 patients given the additional challenges of imaging beyond the usual. The resolution rate of pulmonary thrombi may be relevant for long-term complications of COVID-19 patients. Available studies suggest a somewhat lower rate of clot resolution than in non-COVID-19 PE. Nonetheless, the evidence is of poor quality and too insufficient to draw conclusions from.

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
