# Peer review of "Diagnostic Management of Acute Pulmonary Embolism in COVID-19 and Other Special Patient Populations"

_diagnostics, 2022, doi:10.3390/diagnostics12061350_

Round 1

Reviewer 1 Report

This article describes the application of the diagnosis of suspected PE in COVID-19. The narrative is rigorous and coherent. It gives us a better understanding of the diagnosis of suspected PE, especially in patients with COVID-19.  All in all, this is an article worth reading deeply.  Of course, if the article can provide some strong evidence to support, the quality of the article will be better. 

Author Response

We thank The Reviewer for these kind remarks.
Since unfortunately the available evidence on this topic is very little, we summarized literature representing evidence of best available quality. A re-run of our search string for the period after February 2022 did not result in additional eligible articles.
We have carefully re-read our manuscript for English inaccuracies and have made corrections where needed.

Reviewer 2 Report

The manuscript reviewed discusses highly important problem of LE and COVID-19. 

The major issue is the lack of practical infomation and very short description of existing data about LE and COVID-19. The manuscript contains only description of results without any procedures for the estimation of articles. It is not easy to undestand what kind of reviews the authors presented. 

Author Response

Dear Reviewer, thank you for reviewing our manuscript.
We like to recall that our manuscript concerns a narrative review and not a systematic review or meta-analysis. For this reason we don’t provide a full quality assessment of the discussed studies.
Still, we have made a few revisions, specifying the type of study described.

We would like to respond to the score given to the item: ‘is the work scientifically sound and not misleading’. We believe that our manuscript does not provide misleading evidence because of the following reasons:
- We made a careful selection of relevant studies obtained through our extensive search string. 
- We decided not to focus on (mostly poorly performed) retrospective studies solely focusing on the optimal cut-off between sensitivity and specificity on the ROC curve for D-dimer levels, as this generally results in unacceptable proportion of missed PE due to suboptimal values for both sensitivity as specificity. Notably, none of these thresholds have been studied prospectively. 
We unfortunately have to conclude that available evidence is scarce and of poor quality.

Reviewer 3 Report

It is advisable to extend the bibliography search to be included in the narrative review, also in consideration of the specific topic.

Author Response

Dear Reviewer, thank you for reviewing our manuscript and your suggestion.

We performed an extensive literature search with the help of an experienced information specialist. As it does not concern a systematic review, we believe our thorough bibliography search (as listed below) is sufficient. In addition to this, we have re-run our search string for the period between February-2022 and today. Nevertheless, this did not result in any additional eligible articles. Of note, the available evidence on this topic is unfortunately little and of low quality. We decided to not focus on less relevant details of poorly performed retrospective studies.

Appendix A

The following databases were searched (09-02-2022):

  • Pubmed
  • Embase, including meeting abstract references
  • Web of Science
  • Cochrane Library
  • Emcare
  • WHO COVID-19 Database, including meeting abstract references
  • COVID-19 Evidence
  • Google Scolar

Search string:
PubMed

http://www.ncbi.nlm.nih.gov/pubmed?otool=leiden

(("diagnostic algorithm"[tw] OR "diagnostic algorithms"[tw] OR "diagnostic algorithm*"[tw] OR "diagnosis algorithm"[tw] OR "diagnosis algorithms"[tw] OR "diagnosis algorithm*"[tw] OR (("Algorithms"[mesh] OR "algorithm"[tw] OR "algorithms"[tw] OR "algorithm*"[tw]) AND ("Diagnosis"[mesh] OR "diagnosis"[subheading] OR "diagnosis"[tw] OR "diagnostic"[tw] OR "diagnos*"[tw])) OR "diagnostic strategy"[tw] OR "diagnostic strategies"[tw] OR "rule out"[tw] OR "ruling out"[tw] OR "ruled out"[tw]) AND ("Pulmonary Embolism"[Mesh] OR "pulmonary embolism"[tw] OR "pulmonary embol*"[tw] OR "pulmonary thromboembolism"[tw] OR "pulmonary thromboembol*"[tw] OR "pulmonary thrombo embolism"[tw] OR "pulmonary thrombo embol*"[tw] OR "lung embolism"[tw] OR "lung embol*"[tw] OR "lung thromboembolism"[tw] OR "lung thromboembol*"[tw] OR "Venous Thromboembolism"[Mesh] OR "Venous Thromboembolism"[tw] OR "Thrombosis"[mesh] OR "thrombosis"[tw]) AND ("SARS-CoV-2"[Mesh] OR "COVID-19"[Mesh] OR "COVID-19"[mesh] OR "2019 nCoV Disease"[all fields] OR "2019 nCoV Diseases"[all fields] OR "2019 nCoV Infection"[all fields] OR "2019 nCoV Infections"[all fields] OR "2019 ncov"[all fields] OR "2019 Novel Coronavirus Disease"[all fields] OR "2019 Novel Coronavirus Diseases"[all fields] OR "2019 Novel Coronavirus Infection"[all fields] OR "2019 Novel Coronavirus Infections"[all fields] OR "2019 Novel Coronavirus"[all fields] OR "2019 Novel Coronaviruses"[all fields] OR "2019-nCoV Disease"[all fields] OR "2019-nCoV Diseases"[all fields] OR "2019-nCoV Infection"[all fields] OR "2019-nCoV Infections"[all fields] OR "2019ncov"[all fields] OR "2019-nCoV"[all fields] OR "Coronavirus Disease 19"[all fields] OR "Coronavirus Disease 2019 Virus"[all fields] OR "Coronavirus Disease 2019"[all fields] OR "Coronavirus Disease 2019"[all fields] OR "Coronavirus Disease-19"[all fields] OR "COVID 19 Pandemic"[all fields] OR "COVID 19 Virus Disease"[all fields] OR "COVID 19 Virus Diseases"[all fields] OR "COVID 19 Virus Infection"[all fields] OR "COVID 19 Virus Infections"[all fields] OR "COVID 19 Virus"[all fields] OR "COVID 19 Viruses"[all fields] OR "COVID 19"[all fields] OR "COVID 2019"[all fields] OR "COVID 2019"[all fields] OR "COVID-19 Pandemic"[all fields] OR "COVID-19 Pandemics"[all fields] OR "COVID-19 Virus Disease"[all fields] OR "COVID-19 Virus Diseases"[all fields] OR "COVID-19 Virus Infection"[all fields] OR "COVID-19 Virus Infections"[all fields] OR "COVID-19 Virus"[all fields] OR "COVID-19 Viruses"[all fields] OR "COVID19"[all fields] OR "COVID-19"[all fields] OR "COVID2019"[all fields] OR "ncov 2019"[all fields] OR "ncov2019"[all fields] OR "SARS 2"[all fields] OR "SARS corona virus 2"[all fields] OR "SARS Coronavirus 2 Infection"[all fields] OR "SARS Coronavirus 2 Infections"[all fields] OR "SARS Coronavirus 2"[all fields] OR "SARS CoV 2 Infection"[all fields] OR "SARS CoV 2 Infections"[all fields] OR "SARS CoV 2 Virus"[all fields] OR "SARS CoV 2 Viruses"[all fields] OR "SARS cov 2"[all fields] OR "SARS cov2"[all fields] OR "SARS2"[all fields] OR "SARSCOV 2"[all fields] OR "SARS-COV 2"[all fields] OR "SARS-CoV-2 Infection"[all fields] OR "SARS-CoV-2 Infections"[all fields] OR "SARS-CoV-2 Virus"[all fields] OR "SARS-CoV-2 Viruses"[all fields] OR "SARSCOV2"[all fields] OR "SARS-COV2"[all fields] OR "Severe Acute Respiratory Syndrome Coronavirus 2"[all fields] OR "severe acute respiratory syndrome cov 2"[all fields] OR "severe acute respiratory syndrome cov2"[all fields] OR "Wuhan Coronavirus"[all fields] OR "Wuhan Seafood Market Pneumonia Virus"[all fields] OR (("novel coronavirus*"[all fields] OR "novel corona virus*"[all fields] OR "new coronavirus*"[all fields] OR "new corona virus*"[all fields] OR (("coronavirus*"[all fields] OR "corona virus*"[all fields] OR "pneumonia virus*"[all fields] OR "cov"[all fields] OR "ncov"[all fields]) AND ("outbreak"[all fields] OR "wuhan"[all fields] OR "new"[all fields] OR "novel"[all fields] OR "2019"[tw]))) AND ("2019/01/01"[PDAT] : "3000/12/31"[PDAT])) OR "COVID-19 Serological Testing"[Mesh] OR "3C-like proteinase, SARS-CoV-2"[Supplementary Concept] OR "COVID-19 Vaccines"[Mesh] OR "nucleocapsid phosphoprotein, SARS-CoV-2"[Supplementary Concept] OR "spike protein, SARS-CoV-2"[Supplementary Concept] OR "COVID-19 Testing"[Mesh] OR "SARS-Cov-2 variant VUI-202012-01"[Supplementary Concept] OR "papain-like protease, SARS-CoV-2"[Supplementary Concept] OR "COVID-19 serotherapy"[Supplementary Concept] OR "nidoviral uridylate-specific endoribonuclease"[Supplementary Concept] OR "NSP12 protein, SARS-CoV-2"[Supplementary Concept] OR "PittCoVacc"[Supplementary Concept] OR "NSP16 protein, SARS-CoV-2"[Supplementary Concept] OR "NS8 protein, SARS-CoV-2"[Supplementary Concept] OR "Ns7b protein, SARS-CoV-2"[Supplementary Concept] OR "NSP1 protein, SARS-CoV-2"[Supplementary Concept] OR "pediatric multisystem inflammatory disease, COVID-19 related"[Supplementary Concept] OR "ORF1ab polyprotein, SARS-CoV-2"[Supplementary Concept]) NOT (("Case Reports"[ptyp] OR "case report"[ti]) NOT ("Review"[ptyp] OR "review"[ti])))

AND ("xxxx/01/01"[PDAT] : "3000/12/31"[PDAT])

  • MEDLINE via OVID
    http://gateway.ovid.com/ovidweb.cgi?T=JS&MODE=ovid&NEWS=n&PAGE=main&D=medall
  • MEDLINE via Ebsco
    https://search.ebscohost.com/login.aspx?authtype=ip,cookie,url,uid&groupid=main&profile=ehost&defaultdb=mdc
  • MEDLINE via Web of Science
    http://apps.webofknowledge.com/MEDLINE_GeneralSearch_input.do?product=MEDLINE&SID=D2KPioavtVCTj1ZPoSq&search_mode=GeneralSearch

Chrome https://sr-accelerator.com/#/polyglot 

Embase

http://ovidsp.ovid.com/ovidweb.cgi?T=JS&PAGE=main&MODE=ovid&D=oemezd
(("diagnostic algorithm".mp OR "diagnostic algorithms".mp OR "diagnostic algorithm*".mp OR "diagnosis algorithm".mp OR "diagnosis algorithms".mp OR "diagnosis algorithm*".mp OR ((exp *"Algorithm"/ OR "algorithm".ti,ab OR "algorithms".ti,ab OR "algorithm*".ti,ab) AND (exp *"Diagnosis"/ OR "diagnosis".fs OR "diagnosis".ti,ab OR "diagnostic".ti,ab OR "diagnos*".ti,ab)) OR "diagnostic strategy".ti,ab OR "diagnostic strategies".ti,ab OR "rule out".ti,ab OR "ruling out".ti,ab OR "ruled out".ti,ab) AND (exp *"Lung Embolism"/ OR "pulmonary embolism".ti,ab OR "pulmonary embol*".ti,ab OR "pulmonary thromboembolism".ti,ab OR "pulmonary thromboembol*".ti,ab OR "pulmonary thrombo embolism".ti,ab OR "pulmonary thrombo embol*".ti,ab OR "lung embolism".ti,ab OR "lung embol*".ti,ab OR "lung thromboembolism".ti,ab OR "lung thromboembol*".ti,ab OR exp *"Venous Thromboembolism"/ OR "Venous Thromboembolism".ti,ab OR exp *"Thrombosis"/ OR "thrombosis".ti,ab) AND (SARS coronavirus/ OR ("COVID-19" OR "severe acute respiratory syndrome coronavirus 2" OR 2019ncov OR "2019 ncov" OR novel coronavirus* OR novel corona virus* OR ((coronavirus* OR corona virus* OR pneumonia virus* OR cov OR ncov) AND (outbreak OR wuhan OR "new")) OR covid19 OR "covid 19" OR ((coronavirus* OR corona virus*) AND 2019) OR "sars cov 2" OR sars2 OR new coronavirus* OR new corona virus* OR "ncov 2019" OR "sars coronavirus 2" OR "sars corona virus 2" OR "severe acute respiratory syndrome cov 2" OR "severe acute respiratory syndrome cov2" OR "COVID-19" OR "COVID19" OR "COVID2019" OR "COVID 2019" OR "severe acute respiratory syndrome coronavirus 2" OR SARS-COV* OR SARSCOV*).af) AND 2019:3000.(sa_year) NOT (("Case Report"/ OR "case report".ti) NOT (exp "Review"/ OR "review".ti)))

o             NOT conference review.pt

o             NOT (conference review or conference abstract).pt

o             AND (conference abstract).pt

AND xxxx:2023.(sa_year)

Web of Science

http://isiknowledge.com/wos   

((TS=("diagnostic algorithm" OR "diagnostic algorithms" OR "diagnostic algorithm*" OR "diagnosis algorithm" OR "diagnosis algorithms" OR "diagnosis algorithm*") OR TI=(("Algorithm" OR "algorithm" OR "algorithms" OR "algorithm*") AND ("diagnosis" OR "diagnostic" OR "diagnos*")) OR AK=(("Algorithm" OR "algorithm" OR "algorithms" OR "algorithm*") AND ("diagnosis" OR "diagnostic" OR "diagnos*")) OR AB=(("Algorithm" OR "algorithm" OR "algorithms" OR "algorithm*") AND ("diagnosis" OR "diagnostic" OR "diagnos*")) OR TS=("diagnostic strategy" OR "diagnostic strategies" OR "rule out" OR "ruling out" OR "ruled out")) AND TS=("Lung Embolism" OR "pulmonary embolism" OR "pulmonary embol*" OR "pulmonary thromboembolism" OR "pulmonary thromboembol*" OR "pulmonary thrombo embolism" OR "pulmonary thrombo embol*" OR "lung embolism" OR "lung embol*" OR "lung thromboembolism" OR "lung thromboembol*" OR "Venous Thromboembolism" OR "Venous Thromboembolism" OR "Thrombosis" OR "thrombosis") AND TS=("COVID-19" OR "severe acute respiratory syndrome coronavirus 2" OR 2019ncov OR "2019 ncov" OR novel coronavirus* OR novel corona virus* OR ((coronavirus* OR corona virus* OR pneumonia virus* OR cov OR ncov) AND (outbreak OR wuhan OR "new")) OR covid19 OR "covid 19" OR ((coronavirus* OR corona virus*) AND 2019) OR "sars cov 2" OR sars2 OR new coronavirus* OR new corona virus* OR "ncov 2019" OR "sars coronavirus 2" OR "sars corona virus 2" OR "severe acute respiratory syndrome cov 2" OR "severe acute respiratory syndrome cov2" OR "COVID-19" OR "COVID19" OR "COVID2019" OR "COVID 2019" OR "severe acute respiratory syndrome coronavirus 2" OR SARS-COV* OR SARSCOV*) AND py=(2019 OR 2020 OR 2021 OR 2022 OR 2023) NOT TI=(("Case Report" OR "case") NOT ("review")))

Cochrane

https://www.cochranelibrary.com/advanced-search/search-manager

((("diagnostic algorithm" OR "diagnostic algorithms" OR "diagnostic algorithm*" OR "diagnosis algorithm" OR "diagnosis algorithms" OR "diagnosis algorithm*"):ti,ab,kw OR (("Algorithm" OR "algorithm" OR "algorithms" OR "algorithm*") AND ("diagnosis" OR "diagnostic" OR "diagnos*")):ti,ab,kw OR ("diagnostic strategy" OR "diagnostic strategies" OR "rule out" OR "ruling out" OR "ruled out"):ti,ab,kw) AND ("Lung Embolism" OR "pulmonary embolism" OR "pulmonary embol*" OR "pulmonary thromboembolism" OR "pulmonary thromboembol*" OR "pulmonary thrombo embolism" OR "pulmonary thrombo embol*" OR "lung embolism" OR "lung embol*" OR "lung thromboembolism" OR "lung thromboembol*" OR "Venous Thromboembolism" OR "Venous Thromboembolism" OR "Thrombosis" OR "thrombosis"):ti,ab,kw AND ("COVID 19" OR "severe acute respiratory syndrome coronavirus 2" OR 2019ncov OR "2019 ncov" OR novel coronavirus* OR novel corona virus* OR ((coronavirus* OR corona virus* OR pneumonia virus* OR cov OR ncov) AND (outbreak OR wuhan OR "new")) OR covid19 OR "covid 19" OR ((coronavirus* OR corona virus*) AND 2019) OR "sars cov 2" OR sars2 OR new coronavirus* OR new corona virus* OR "ncov 2019" OR "sars coronavirus 2" OR "sars corona virus 2" OR "severe acute respiratory syndrome cov 2" OR "severe acute respiratory syndrome cov2" OR "COVID 19" OR "COVID19" OR "COVID2019" OR "COVID 2019" OR "severe acute respiratory syndrome coronavirus 2" OR SARS COV* OR SARSCOV*):ti,ab,kw)

Emcare
http://ovidsp.ovid.com/ovidweb.cgi?T=JS&NEWS=n&CSC=Y&PAGE=main&D=emcr

(("diagnostic algorithm".mp OR "diagnostic algorithms".mp OR "diagnostic algorithm*".mp OR "diagnosis algorithm".mp OR "diagnosis algorithms".mp OR "diagnosis algorithm*".mp OR ((exp *"Algorithm"/ OR "algorithm".ti,ab OR "algorithms".ti,ab OR "algorithm*".ti,ab) AND (exp *"Diagnosis"/ OR "diagnosis".ti,ab OR "diagnostic".ti,ab OR "diagnos*".ti,ab)) OR "diagnostic strategy".ti,ab OR "diagnostic strategies".ti,ab OR "rule out".ti,ab OR "ruling out".ti,ab OR "ruled out".ti,ab) AND (exp *"Lung Embolism"/ OR "pulmonary embolism".ti,ab OR "pulmonary embol*".ti,ab OR "pulmonary thromboembolism".ti,ab OR "pulmonary thromboembol*".ti,ab OR "pulmonary thrombo embolism".ti,ab OR "pulmonary thrombo embol*".ti,ab OR "lung embolism".ti,ab OR "lung embol*".ti,ab OR "lung thromboembolism".ti,ab OR "lung thromboembol*".ti,ab OR exp *"Venous Thromboembolism"/ OR "Venous Thromboembolism".ti,ab OR exp *"Thrombosis"/ OR "thrombosis".ti,ab) AND (SARS coronavirus/ OR ("COVID-19" OR "severe acute respiratory syndrome coronavirus 2" OR 2019ncov OR "2019 ncov" OR novel coronavirus* OR novel corona virus* OR ((coronavirus* OR corona virus* OR pneumonia virus* OR cov OR ncov) AND (outbreak OR wuhan OR "new")) OR covid19 OR "covid 19" OR ((coronavirus* OR corona virus*) AND 2019) OR "sars cov 2" OR sars2 OR new coronavirus* OR new corona virus* OR "ncov 2019" OR "sars coronavirus 2" OR "sars corona virus 2" OR "severe acute respiratory syndrome cov 2" OR "severe acute respiratory syndrome cov2" OR "COVID-19" OR "COVID19" OR "COVID2019" OR "COVID 2019" OR "severe acute respiratory syndrome coronavirus 2" OR SARS-COV* OR SARSCOV*).af) AND 2019:3000.(sa_year) NOT (("Case Report"/ OR "case report".ti) NOT (exp "Review"/ OR "review".ti)))

WHO Covid-19 database

https://search.bvsalud.org/global-literature-on-novel-coronavirus-2019-ncov/

(("diagnostic algorithm" OR "diagnostic algorithms" OR "diagnosis algorithm" OR "diagnosis algorithms" OR (("algorithm" OR "algorithms") AND ("diagnosis" OR "diagnostic")) OR "diagnostic strategy" OR "diagnostic strategies" OR "rule out" OR "ruling out" OR "ruled out") AND ("Lung Embolism" OR "pulmonary embolism" OR "pulmonary thromboembolism" OR "pulmonary thrombo embolism" OR "lung embolism" OR "lung thromboembolism" OR "Venous Thromboembolism" OR "thrombosis"))

COVID-19 Evidence

https://app.iloveevidence.com/loves/5e6fdb9669c00e4ac072701d?utm=aile

(("diagnostic algorithm" OR "diagnostic algorithms" OR "diagnosis algorithm" OR "diagnosis algorithms" OR (("algorithm" OR "algorithms") AND ("diagnosis" OR "diagnostic")) OR "diagnostic strategy" OR "diagnostic strategies" OR "rule out" OR "ruling out" OR "ruled out") AND ("Lung Embolism" OR "pulmonary embolism" OR "pulmonary thromboembolism" OR "pulmonary thrombo embolism" OR "lung embolism" OR "lung thromboembolism" OR "Venous Thromboembolism" OR "thrombosis"))

Google Scholar

https://scholar.google.com

Round 2

Reviewer 2 Report

Dear authors, I have no critical comments.